# BCG Vaccination and Mortality of COVID-19 across 173 Countries: An Ecological Study

**DOI:** 10.3390/ijerph17155589

**Published:** 2020-08-03

**Authors:** Mitsuyoshi Urashima, Katharina Otani, Yasutaka Hasegawa, Taisuke Akutsu

**Affiliations:** 1Division of Molecular Epidemiology, The Jikei University School of Medicine, Tokyo 105-8461, Japan; katharina@jikei.ac.jp (K.O.); yhearth102523@jikei.ac.jp (Y.H.); t-akutsu@jikei.ac.jp (T.A.); 2Advanced Therapies Innovation Department, Siemens Healthcare K.K., Tokyo 141-8644, Japan; 3Hitachi, Ltd. Research & Development Group, Tokyo 185-8601, Japan

**Keywords:** urbanization, Bacillus Calmette–Guérin, BCG, vaccination, coronavirus disease 2019, COVID-19, SARS-CoV-2, ecological study, morbidity, mortality

## Abstract

Ecological studies have suggested fewer COVID-19 morbidities and mortalities in Bacillus Calmette–Guérin (BCG)-vaccinated countries than BCG-non-vaccinated countries. However, these studies obtained data during the early phase of the pandemic and did not adjust for potential confounders, including PCR-test numbers per population (PCR-tests). Currently—more than four months after declaration of the pandemic—the BCG-hypothesis needs reexamining. An ecological study was conducted by obtaining data of 61 factors in 173 countries, including BCG vaccine coverage (%), using morbidity and mortality as outcomes, obtained from open resources. ‘Urban population (%)’ and ‘insufficient physical activity (%)’ in each country was positively associated with morbidity, but not mortality, after adjustment for PCR-tests. On the other hand, recent BCG vaccine coverage (%) was negatively associated with mortality, but not morbidity, even with adjustment for percentage of the population ≥ 60 years of age, morbidity, PCR-tests and other factors. The results of this study generated a hypothesis that a national BCG vaccination program seems to be associated with reduced mortality of COVID-19, although this needs to be further examined and proved by randomized clinical trials.

## 1. Introduction

Currently, more than four months since declaration of the coronavirus disease 2019 (COVID-19) as a pandemic by the World Health Organization (WHO) on March 11th, 2020, more than 14 million people have been infected with the virus and more than half a million have died worldwide. Marked differences in COVID-19 mortalities have been observed in different countries. For example, the mortality per million population is till now several tens of times or even higher in Western countries, e.g., Belgium (845), the United Kingdom (UK, 664), the United States of America (USA, 426) and Germany (109), than in Asian countries, e.g., India (19), Japan (8) and China (3), as of 17 July 2020. This is quite the opposite of what was reported during the 1918–20 influenza pandemic, the so called Spanish flu, in which the population mortality was over 30-fold higher, with excess death rates observed in low-income countries, such as India, than in high-income countries, such as those in the West [1]. Higher morbidities and mortalities may be explained by easy access to diagnostic polymerase chain reaction tests (PCR-tests) for severe acute respiratory syndrome coronavirus 2 (SARS-CoV-2) in these Western countries. In contrast, they may be underestimated in middle- and low-income countries due to low capacities of PCR-testing and poor access to health care. Alternatively, there is growing concern that racial and ethnic minority, as well as socioeconomic and biological factors may influence the high morbidity and mortality. A retrospective study of integrated health care systems suggested that Black race compared with White race, increasing age, male sex, having a greater burden of comorbidities, type of public insurance, residence in a low-income area and obesity were associated with increased odds of hospital admission [2]. On the other hand risk of in-hospital mortality increased only in aged patients and was not associated with the Black race, sex, comorbidities, obesity and other factors after multivariate adjustment [2]; this phenomenon was also confirmed in other races, i.e., Asians and Hispanics, compared with the White race [3,4]. Moreover, since COVID-19 is an infectious disease that spreads mainly through the droplet route by close contact in dense human societies, metropolitan areas, such as New York City in the USA [5] and Lombardy in Italy [6], Paris in France [7], Sao Paulo in Brazil [8], and so on, have tended to be regional epicenters. However, associations between population dynamics, e.g., population size, density, migrants and urbanization and the morbidity/mortality of COVID-19 have not yet been well examined.

In addition to the factors listed above, several scientists proposed a hypothesis [9,10,11,12] that Bacillus Calmette–Guérin (BCG) vaccination has preventive effects not only against tuberculosis, i.e., the target disease of the vaccine, but also other non-specific infectious diseases, i.e., off-target diseases such as COVID-19, through epigenetic mechanisms [13,14,15]. In fact, ecological studies have suggested that both COVID-19 morbidities and mortalities were less in BCG-vaccinated countries than in BCG-non-vaccinated countries [16,17,18]. However, because these studies only obtained data during the early phase of the pandemic, by which time the disease load had escalated in Western countries, but not yet escalated in low- and middle-income countries where BCG is given at birth and did not adjust for any potential confounders, including PCR-test number per million population, the BCG-hypothesis needs to be reexamined now, more than four months after declaration of the pandemic, since the number of PCR-confirmed COVID-19 cases are still growing in many countries. We therefore aimed to explore whether recent BCG vaccine coverage is associated with COVID-19 morbidity and/or mortality rates, using linear regression models to explore associations between the two continuous random variables adjusted for a variety of potential confounders, such as median age and body mass index (BMI) in individual countries through this ecological study.

## 2. Materials and Methods

### 2.1. Ethics Statement

Institutional review board approval for this work was not sought due to the use of publicly available data obtained from open resources.

### 2.2. Study Design

This ecological study compared population data of each country.

### 2.3. Data Resources

All data were obtained from open resources: Information on total number of cases, total number of deaths and number of PCR-tests performed per million population were obtained from ‘Coronavirus Update’ [19], data regarding population dynamics were obtained from ‘World Population Clock’ [20], socioeconomic covariates were obtained from ‘the United Nations database’ [21], BCG vaccine coverage data were obtained from ‘The BCG World Atlas’ [22] and ‘The global health observatory’ on the WHO homepage [18] and other health related data were from the same WHO site [23]. Definitions of each of the covariates are described in the Supplement. Only countries that had data of both total deaths and BCG vaccine coverage were included for analyses in this study.

### 2.4. Outcomes and PCR-Tests

The outcomes evaluated in this study were COVID-19 morbidity, i.e., total COVID-19 cases per million population and COVID-19 mortality, i.e., total COVID-19-related deaths per million population, in each country, obtained from ‘Worldometer COVID-19 Data’ on July 17th, 2020 [19]. Data on the total number of PCR-tests performed per million population were simultaneously obtained from the same ‘Worldometer COVID-19 Data’ website. PCR-test positivity was simply calculated as the total COVID-19 case number divided by the total number of PCR-tests performed in each country.

### 2.5. BCG Vaccine Coverage

Recent BCG vaccine coverage was defined as the percentage of the vaccinated population among one-year-olds in each country (World Health Data Platform, the World Health Observatory, BCG-Immunization coverage estimates) [23]. For countries that have already stopped a national BCG vaccine immunization program [22], which includes a significant number of countries, their BCG vaccine coverage rate was counted as zero percent in this study.

### 2.6. Sixty-One Covariates

The covariates evaluated in this study, the definitions of which are described in the Appendix A, included: (1) Population (n); (2) yearly change in population (%); (3) net change in population (n); (4) population density (n/km^2^); (5) land area (km^2^); (6) net number of migrants (n); (7) fertility rate (n); (8) median age (years); (9) urban population percentage (%); (10) world share (%); (11) population between age 0 to 14 years (%); (12) population ≥ 60 years of age (%); (13) population ≥ 70 years of age (%); (14) gross domestic product (GDP) (million US dollars); (15) GDP per capita (US dollars); (16) total unemployment rate (%); (17) male unemployment rate (%); (18) female unemployment rate (%); (19) total labor force participation rate (%); (20) male labor force participation rate (%); (21) female labor force participation rate (%); (22) annual incidence of tuberculosis per 100,000 population (n); (23) international health regulation (IHR) score (%); (24) universal health coverage (UHC) index; (25) hospital beds (n) per 10,000 population; (26) medical doctors (n) per 10,000 population; (27) nursing midwifery (n) per 10,000 population; (28) licensed qualified anesthesiologists who usually cover management of intensive care units actively working (n) per 10,000 population; (29) total expenditure on health as a percentage of GDP (%); (30) population with household expenditures on health greater than 10% of total household expenditure or income (%); (31) prevalence of high blood pressure (systolic ≥ 140 or diastolic ≥ 90 mmHg) (%); (32) prevalence of elevated fasting blood glucose levels (≥7.0 mmol/L or on anti-diabetic medication); (33) prevalence of elevated total cholesterol levels (≥5.0 mmol/L) (%); (34) mean BMI (body weight [kg]/height^2^ [m^2^]); (35) prevalence of obesity among adults, BMI ≥ 30 kg/m^2^ (%); (36) prevalence of ‘overweight’ people among adults, BMI ≥ 25 kg/m^2^ (%); (37) alcohol drinking, total per capita (≥15 years of age) consumption (in liters of pure alcohol over a calendar year); (38) prevalence of smoking any tobacco product among males aged ≥ 15 years (%); (39) prevalence of smoking any tobacco product among females aged ≥ 15 years (%); (40) prevalence of insufficient physical activity among adults aged ≥ 18 years (%); (41) estimated population-based prevalence of depression (%); (42) neonatal mortality rate (n per 1000 live births); (43) infantile mortality rate (n per 1000 live births); (44) under-five mortality rate (probability of dying by the age of 5 years per 1000 live births); (45) mortality rate for 5–14-year-olds (probability of dying per 1000 children aged 5–14 years); (46) adult mortality rate (probability of dying between 15 and 60 years of age per 1000 population); (47) probability of dying between age 30 and exact age 70 years from any of the following causes: cardiovascular disease, cancer, diabetes or chronic respiratory disease; (48) life expectancy at birth (years); (49) life expectancy at age 60 years (years); (50) healthy life expectancy (HALE) at birth (years); (51) HALE at age 60 years (years); (52) death due to chronic obstructive pulmonary disease (%); (53) death due to ischemic heart disease (%); (54) death due to lower respiratory infections (%); (55) death due to stroke (%); (56) death due to tracheal, bronchial and lung cancers (%); (57) total of (52) to (56) as ambient and household air pollution-attributable death rate (n per 100,000 population); and (58) annual mean concentration of particulate matter less than 2.5 microns in diameter (PM_2.5_) [µg/m^3^] in urban areas; and (59) coverage rate with the first dose of a measles-containing-vaccine (MCV1) among one-year-olds (%) as well as (60) recent BCG coverage and (61) PCR-tests number.

### 2.7. Statistics

#### Linear Regression Models

For preprocessed data, outcomes, i.e., morbidity and mortality per million population were transformed to the common logarithm (log10) to adjust for normality of the distribution, which was verified by means of kurtosis tests. When the number of total deaths was zero, these were changed to 0.01 per million population, because zero cannot be transformed to the common logarithm. Variance inflation factor (VIF) was used to detect the presence of multicollinearity. Only one variable among biologically similar variables, e.g., ‘median age (years)’, ‘≥ 60 years of age (%)’ and ‘≥ 70 years of age’, was selected to maximize adjusted R^2^ in the multi-linear regression models to avoid the influence of collinearity. If the variance inflation factor (VIF) of certain covariates was more than 10, then the covariates were avoided in multivariate analyses because of a collinearity issue. Multiple linear regression models were used to screen potential risk or preventive factors associated with morbidity by adjusting for PCR-test numbers transformed to the common logarithm (log10) and those associated with mortality were screened by adjusting for PCR-test numbers and morbidity per million population. Considering type I error due to a multiple testing, the significance level of alfa was set as *p* < 0.001. Then, all the screened factors were assessed in a multi-linear regression model to determine significant factors with *p* < 0.05 as the cutoff point. Each model was evaluated by adjusted R^2^ as a coefficient of determination. Pearson’s correlation coefficient for variables with normal distributions or Spearman’s rank correlation for variables with non-normal distributions, represented as rho, was used to quantify the strengths of associations between morbidity, mortality and significant factors determined by the final models, as: absolute value of rho ≥ 0.5: very strong; rho ≥ 0.4: strong; 0.4 > rho ≥ 0.2: moderate; and rho < 0.2: weak associations. Data were analyzed using Stata version 14.0 software (StataCorp LP, College Station, TX, USA).

## 3. Results

### 3.1. Variability of COVID-19 Morbidity and Mortality across 173 Countries

A total of 173 countries that had data of both total COVID-19 deaths and BCG vaccine coverage were included for analyses in this study. The 20 countries with highest and lowest COVID-19 morbidities (Table 1) and mortalities (Table 2), as well as their PCR-test rate per million population are shown below. Marked differences in morbidities and mortalities were observed among these countries, ranging from 1 (Papua New Guinea) to 37,566 (Qatar) and from 0 (Vietnam, etc.) to 845 (Belgium), respectively. Six and thirteen countries that do not have a BCG national vaccine program at present (indicated with bold and mark “∫”) were included in the 20 countries with the highest COVID-19 morbidity and mortality, respectively.

Histograms of morbidities and mortalities were drawn as normal density plots (Figure 1). Although the histograms of morbidity and mortality were skewed to the right, they followed a normal distribution by transformation with the common logarithm (log10).

### 3.2. Associations between Morbidity, Mortality and PCR-Tests per Million Population

First, associations represented by rho and VIF between morbidity (*n* = 173), mortality (*n* = 173) and PCR-tests (*n* = 155) are shown (Figure 2A). These three variables were predicted to have very strong and positive associations with each other. However, VIFs were less than 2 among these three factors. Multicollinearity is considered to be present when the VIF is higher than 5 to 10 [24]. Thus, any variable with a VIF < 5.0 was considered for inclusion in multiple linear regression analyses. Considering the number of PCR-tests per million population may exhibit associations with morbidity, adjustment was performed for the PCR-test number in every analysis when screening for the risk factors of COVID-19 morbidity per million population (Figure 2B). Considering morbidities and the number of PCR-tests per million population may exhibit associations with mortality, adjustment was performed for the morbidity and PCR-test number in every analysis when screening for the risk factors of COVID-19 mortality per million population (Figure 2C).

Since fewer PCR-tests may underestimate morbidity and mortality, the association between mortality as the outcome and morbidity as the exposure was adjusted for number of PCR-tests performed (Table 3). In this multiple regression analysis, higher morbidity was associated with higher mortality, whereas more PCR-tests were associated with lower mortality.

Evaluation of the association between PCR-test positivity and mortality, shown as a scatter plot, indicated a very strong association between them (rho = 0.54) (Figure 3). Countries with higher PCR-positivity rates tended to have higher mortality rates. PCR-test positivity rates of countries where no deaths due to COVID-19 were observed were less than 3.5%. Minimum positivity and no deaths were observed in Vietnam.

### 3.3. Screening Factors Associated with Morbidity

Among the 59 covariates, plus BCG vaccine coverage and PCR-testing rate, i.e., a total of 61 factors, ‘urban population’ and ‘insufficient physical activity’ were significantly (*p* < 0.001) associated with morbidity after adjustment for PCR-test rate (Table 4). Next, these two significant factors were used in a multi-linear regression model to eliminate confounding (Table 5). As a result, ‘urban population’ (*p* = 0.02) and ‘insufficient physical activity’ (*p* = 0.01) remained significant factors associated with COVID-19 morbidity, even after adjustment for PCR-tests. The adjusted R^2^ was 0.5037.

The association between ’urban population’ and morbidity, demonstrated below as a scatter plot, showed a very strong association (rho = 0.55) (Figure 4).

The association between ‘insufficient physical activity’ and morbidity, demonstrated below as a scatter plot, also showed a very strong association (rho = 0.52) (Figure 5).

COVID-19-related morbidity rates per million population on July 17th were transformed to the common logarithm (log10) in the graph. Countries that had never had or that had stopped a national program of BCG vaccination are indicated in red, while countries that currently follow a national BCG vaccine program are indicated in black. Selected country’s name was shown using three-letter country codes.

### 3.4. Screening Factors Associated with Mortality

Among the 58 covariates evaluated, plus BCG vaccine coverage, adjusted for morbidity and PCR-tests, age-related factors, i.e., median age, ≥60 years of age (%) and ≥70 years of age (%), were significantly (*p* < 0.001) associated with mortality (Table 4). Since these three age-related factors had collinearity for mortality, ‘≥ 60 years of age’ was selected to maximize adjusted R^2^ of the multi-linear regression models. Moreover, ‘BCG vaccine coverage’, ‘Elevated total cholesterol levels’ and ‘Life expectancy at 60 years of age’, were also significant (*p* < 0.001) factors associated with mortality. Next, these significant factors were used in a multi-linear regression model to eliminate confounding (Table 6). As a result, ‘≥60 years of age’ (*p* < 0.001) and ‘BCG vaccine coverage’ (*p* = 0.002) remained significant factors associated with COVID-19 mortality, even after adjustment for morbidity and PCR-tests. The adjusted R^2^ was 0.8254.

Evaluation of the association between ’median age’ and mortality, demonstrated as a scatter plot, showed a very strong association between these two variables (rho = 0.54) (Figure 6). Countries with a larger population ≥ 60 years of age (%) showed a tendency toward a higher mortality rate.

Finally, evaluation of the association between ‘BCG vaccine coverage’ and mortality, demonstrated as a scatter plot, indicated a moderately negative association (rho = −0.29) (Figure 7). Countries with higher BCG vaccine coverage showed a tendency toward lower mortality. Additionally, COVID-19 mortality rates did not have significant associations with BCG strain, e.g., Tokyo 172. Moreover, there were no significant associations between mortality and either the year of stopping or introducing a national BCG vaccine program (data not shown).

## 4. Discussion

Among the variety of parameters abstracted from open resources, ‘BCG vaccine coverage’ had a significant association with COVID-19 mortality, even after adjusting for morbidity, PCR-tests, age, universal health coverage, numbers of medical doctors, elevated total cholesterol and healthy life expectancy. On the other hand, BCG vaccination was not associated with COVID-19 morbidity. The main results of this study are consistent with a very recent article demonstrating that every 10% increase in the BCG index was associated with a 10% reduction in COVID-19 mortality [25]. Moreover, a retrospective cohort study suggested that BCG-vaccinated individuals were less likely to require hospital admission during the disease course [26]. In contrast to BCG, coverage of the measles vaccine, which is also considered to induce heterologous protection against infections through long-term boosting of innate immune responses [9], showed no association with the morbidity and mortality of COVID-19, which was also consistent with a very recent article showing a significant low risk of COVID-19 mortality in countries with higher BCG vaccine coverage, but not with measles vaccine coverage [27].

Moreover, SARS-CoV-2 is a single-stranded positive-sense RNA virus and the BCG vaccine has been shown in controlled trials to reduce the severity of infections by other viruses with such a structure [9]. For example, the BCG vaccine reduced yellow fever vaccine viremia by 71% in volunteers in the Netherlands [28]. However, some countries with a current national BCG vaccination policy have high mortality rates. Plausible reasons for this discrepancy may be: (1) low coverage of BCG vaccination in these countries (% coverage–mortality per million population), e.g., Ireland (18%—354); Portugal (32%—165); and Greece (50%—19): (2) late introduction of BCG vaccine program (year of introduction–mortality per million population), e.g., Iran, (1984–162): (3) oral delivery of the BCG vaccine, e.g., Brazil retained oral delivery of the vaccine until 1977: (4) UK administered the vaccine to older children (12 to 13 years of age) and (5) connection with endemic country by land, e.g., Mexico, Panama, Peru and Chile.

Higher morbidity, but fewer PCR-tests, were associated with higher mortality. Moreover, higher PCR-test positivity was associated with higher mortality, which was consistent with the report by Hisaka et al. [29]. Expanding application of PCR-tests not only to typical symptomatic cases, but also to mild or asymptomatic cases and to those who had close contact with patients, may decrease the PCR-test positivity rate. Thus, enhancing the capacity of PCR-testing may enable identification of cases, so that appropriate measures can be taken to prevent them spreading SARS-CoV-2 to others at home, at their work place or at events of mass gatherings.

In this study, the covariate of ‘urban population’ and ‘insufficient physical inactivity’ had a strong and positive association with morbidity, but not with mortality. People living in urban areas tend to have close contact with a greater number of people per day than those in rural areas, independent of age. A report on 4103 patients with COVID-19 in New York City found that obesity, which may strongly depend on the balance between physical activity and diet, was one of the clinical features leading to hospital admission [30]. On the other hand, older age was associated with higher mortality, which is consistent with previous articles [2,3,4]. From this study, the risk factors for morbidity seem to be different from those associated with mortality, suggesting that factors related to susceptibility may be different from those related to disease severity.

There are several limitations to this study. First, although we selected 61 covariates in this study, we did not evaluate range and timing of non-pharmaceutical interventions, e.g., school closures, workplace closures, cancellation of public events, restrictions on public gatherings, stay-at-home restrictions, restrictions on internal movement, international travel controls, etc., all of which would also have had significant effects on COVID-19-related morbidity and mortality. Therefore, the present study results are burdened with an extreme error. Second, the study design was ecological. Therefore, the outcome of this work should be considered highly limited, with a potential risk of high bias. Consequently, only the hypothesis that BCG vaccination mitigates COVID-19 mortality can be proposed here; cohort or case–control studies and randomized clinical trials, similar to the BCG–CORONA study [31], are required to test this hypothesis. Third, the COVID-19 pandemic is still ongoing, although we have confirmed the results using the latest data. However, the results may be different a few months from now. Fourth, it is clear that an extremely large number of covariates, i.e., 61, were selected for the limited number of 173 countries. This sample size could allow use of a maximum of 10 covariates in multiple regression analysis. Therefore, significant (*p* < 0.001) variables were at first screened after adjustment for PCR-tests and morbidity (Table 4). Then, multivariable linear regression using the screened variables were performed after adjustment for PCR-tests and morbidity (Table 5 for morbidity and Table 6 for mortality). Fifth, mortality in each country should be compared with excess deaths whether these two do not make a big difference. Sixth, the definition of COVID-19 cases may differ by country, e.g., including PCR confirmed, but asymptomatic cases and pneumonia cases with negative PCR-tests. Seventh, there were still residual confounders even after adjustment for PCR-tests.

## 5. Conclusions

Our results suggest the hypothesis that greater BCG vaccine coverage may reduce the risk of deaths due to COVID-19, which needs to be further studied by observational studies and confirmed by randomized clinical trials.

## Figures and Tables

**Figure 1 ijerph-17-05589-f001:**
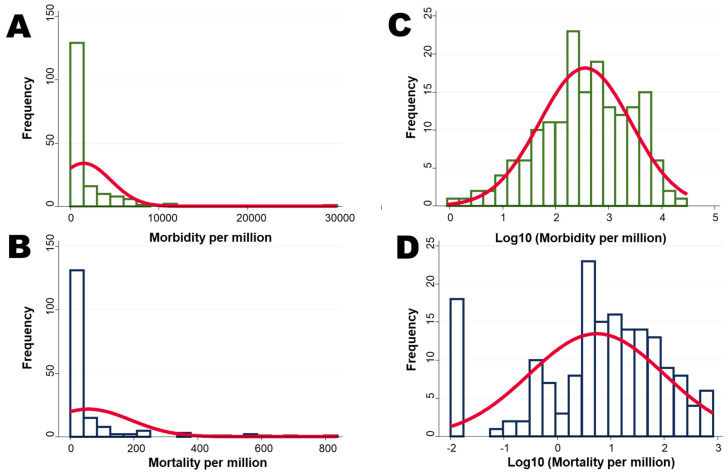
Histograms of morbidities and mortalities drawn as a normal density plot. (**A**) Morbidity per million; (**B**) mortality per million; (**C**) log10 transformed morbidity; and (**D**) log10 transformed mortality.

**Figure 2 ijerph-17-05589-f002:**
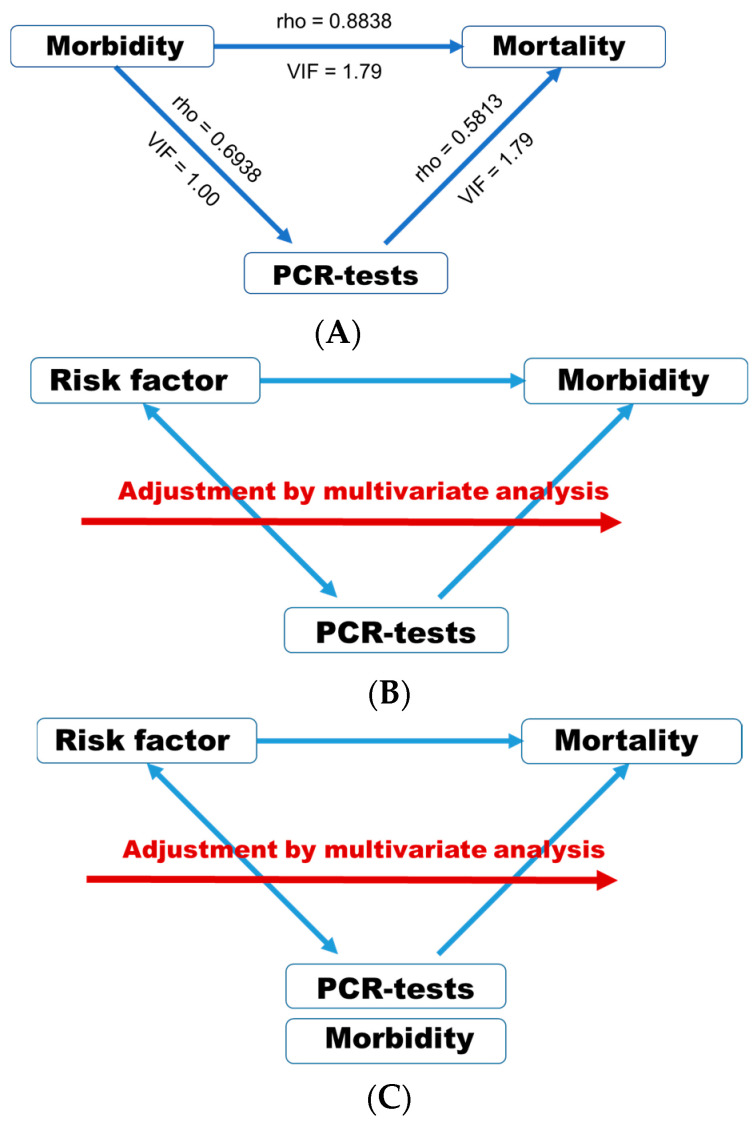
(**A**) Associations between morbidity, mortality and PCR-tests. Either Pearson’s correlation coefficient or Spearman’s rank correlation was applied to calculate rho; (**B**) associations between morbidity and risk factors were adjusted for PCR-tests per million population (log10); (**C**) associations between mortality and risk factors were adjusted for morbidity (log10) and PCR-tests (log10) per million population.

**Figure 3 ijerph-17-05589-f003:**
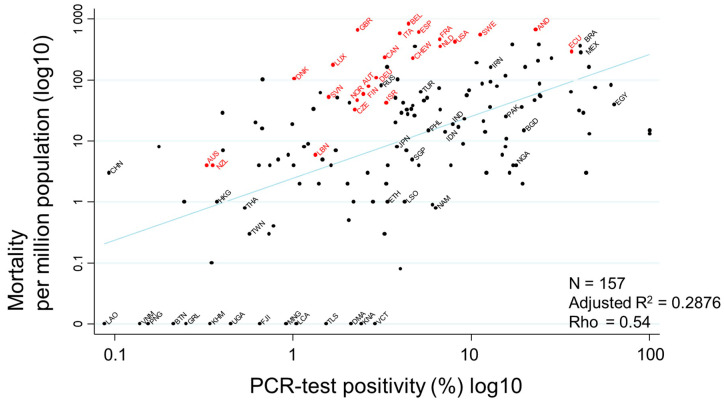
Scatter plot showing the association between PCR-test positivity and mortality. Mortality per million population and PCR-test positivity rates (%) on 17 July 2020 were transformed to the common logarithm (log10) in the graph. Since the variable of ‘PCR-test positivity (log10)’ showed a normal distribution, Pearson’s correlation coefficient was applied to calculate rho, to quantify the strength of the association. Countries that never had or stopped a national BCG vaccine program are indicated in red, while countries with current national BCG vaccine programs are indicated in black. Selected country names are shown using three-letter country codes.

**Figure 4 ijerph-17-05589-f004:**
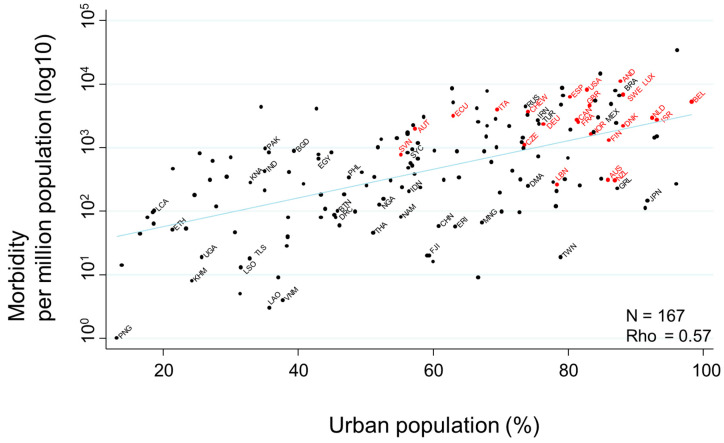
Scatter plot showing the association between urban population and COVID-19 morbidity rates. COVID-19-related morbidity rates per million population on 17 July 2020 were transformed to the common logarithm (log10) in the graph. Since the variable of ‘urban population’ showed a non-normal distribution, Spearman’s rank correlation was applied to calculate rho, to quantify the strength of the association. Countries that had never had or that had stopped a national program of BCG vaccination are indicated in red, while countries that currently follow a national BCG vaccine program are indicated in black. Selected country names are shown using three-letter country codes.

**Figure 5 ijerph-17-05589-f005:**
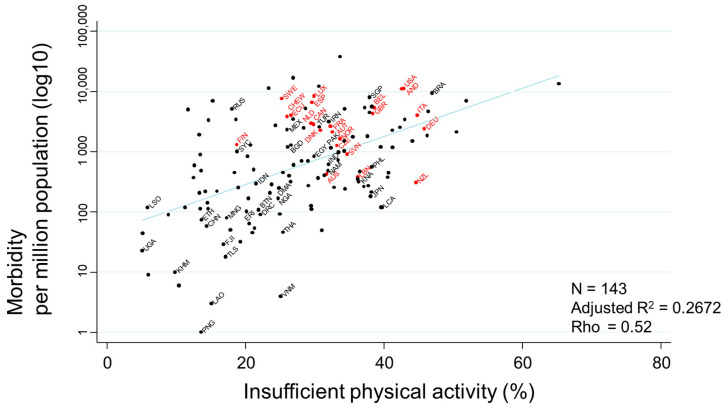
Scatter plot showing the association between insufficient physical activity and COVID-19 morbidity rates.

**Figure 6 ijerph-17-05589-f006:**
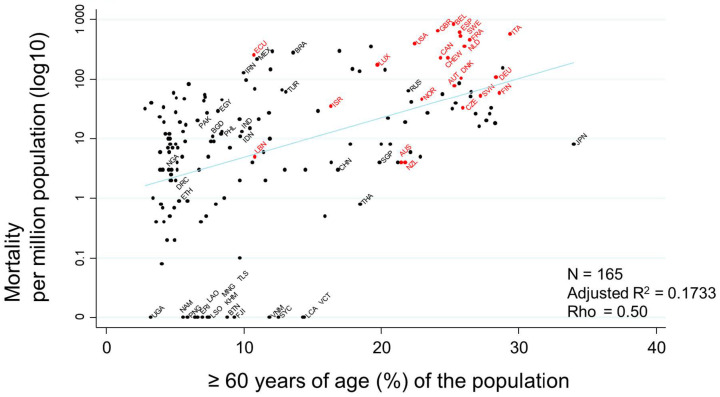
Scatter plot showing the association between ≥60 years of age (%) of the population and mortality rate. Mortalities per million population on 17 July 2020 transformed to the common logarithm (log10) are presented in the graph. Since the variable of ‘percentage of population ≥ 60 years of age (%)’ showed a non-normal distribution, Spearman’s rank correlation was applied to calculate rho, to quantify the strength of the association. Countries that had never had or that had stopped their national BCG vaccine program are indicated in red, while countries that currently follow a national BCG vaccine program are indicated in black. Selected country names are shown using three-letter country codes.

**Figure 7 ijerph-17-05589-f007:**
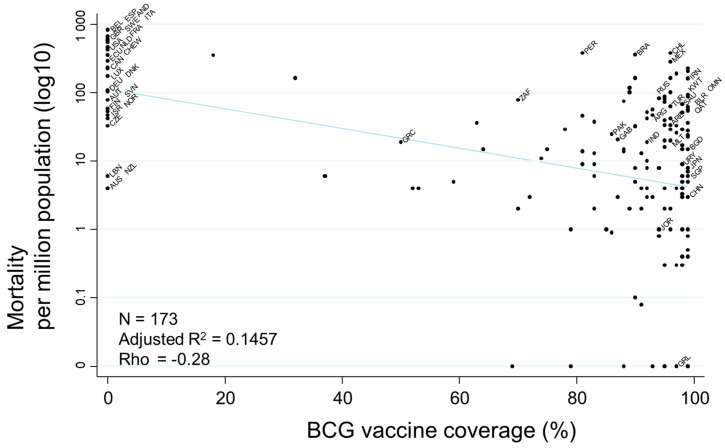
Scatter plot showing the association between COVID-19-related mortality and BCG vaccine coverage. Mortalities per million population on 17 July 2020 are transformed to the common logarithm (log10) in the graph. Since the variable of ‘BCG vaccine coverage’ showed a non-normal distribution, Spearman’s rank correlation was applied to calculate rho, to quantify the strength of the association. Selected country names are shown using three-letter country codes.

**Table 1 ijerph-17-05589-t001:** Twenty countries with the highest and lowest COVID-19 morbidity rates and their PCR-test rate per million population.

The Highest 20 Countries				The Lowest 20 Countries			
Rank	Country	Morbidity *	PCR-Test *	Rank	Country *	Morbidity *	PCR-Test *
1	Qatar	37,566	153,380	173	Papua New Guinea	1	798
2	Chile	16,927	70,696	172	Lao People’s Democratic Republic	3	2990
3	Kuwait	13,496	105,205	171	Vietnam	4	2824
4	Oman	12,244	50,512	170	Myanmar	6	1770
5	Panama	11,406	40,384	169	United Republic of Tanzania	9	
6	Armenia	11,324	47,718	168	Cambodia	10	2896
7	**Andorra ^∫^**	11,156	48,531	167	Angola	18	304
8	**USA ^∫^**	11,118	137,544	166	Timor-Leste	18	1189
9	Peru	10,355	60,747	165	Taiwan	19	3319
10	Brazil	9464	23,098	164	Uganda	23	5133
11	**Luxembourg ^∫^**	8438	502,852	163	Burundi	23	563
12	Singapore	8053	172,506	162	Syrian Arab Republic	27	
13	**Sweden ^∫^**	7610	67,495	161	Fiji	29	4461
14	Saudi Arabia	6983	71,623	160	Gambia	32	1477
15	Belarus	6945	123,003	159	Mozambique	44	1360
16	**Spain ^∫^**	6543	128,893	158	Niger	45	276
17	UAE	5673	436,262	157	Thailand	46	8647
18	South Africa	5464	39,182	156	Burkina Faso	50	
19	**Belgium ^∫^**	5438	121,891	155	Yemen	52	4
20	Maldives	5234	118,769	154	Chad	54	

* Numbers per million population, USA—United States of America, UAE—United Arab Emirates. Bold letters with mark “∫” mean no recent national BCG vaccination program for all.

**Table 2 ijerph-17-05589-t002:** Twenty countries with the highest and lowest COVID-19 mortality rates and their PCR-test rates per million population.

The Highest 20 Countries				The Lowest 20 Countries			
Rank	Country	Mortality *	PCR-Test *	Rank	Country *	Mortality *	PCR-Test *
1	**Belgium ^∫^**	845	121,891	173	Papua New Guinea	0	798
2	**Andorra ^∫^**	673	48,531	172	Lao People’s Democratic Republic	0	2990
3	**UK ^∫^**	664	186,591	171	Vietnam	0	2824
4	Spain	608	128,893	170	Cambodia	0	2896
5	**Italy ^∫^**	579	100,954	169	Timor-Leste	0	1189
6	**Sweden ^∫^**	554	67,495	168	Uganda	0	5133
7	**France ^∫^**	462	39,868	167	Fiji	0	4461
8	**USA ^∫^**	426	137,544	166	Eritrea	0	
9	Peru	382	60,747	165	Mongolia	0	8688
10	Chile	381	70,696	164	Bhutan	0	51,107
11	Brazil	361	23,098	163	Saint Lucia	0	11,494
12	**Netherlands ^∫^**	358	44,588	162	Greenland	0	90,957
13	**Ireland ^∫^**	354	107,707	161	Dominica	0	11,820
14	**Ecuador ^∫^**	295	11,023	160	Saint Vincent and the Grenadines	0	10,932
15	Mexico	286	5958	159	Saint Kitts and Nevis	0	13,248
16	**Canada ^∫^**	234	88,515	158	Seychelles	0	
17	**Switzerland ^∫^**	227	81,825	157	Burundi	0.08	563
18	Panama	227	40,384	156	Myanmar	0.1	1770
19	Armenia	205	47,718	155	Taiwan	0.3	3319
20	North Macedonia	192	38,771	154	Mozambique	0.3	1360

* Numbers per million population, USA—United States of America, UK—United Kingdom, VCT—Saint Vincent and the Grenadines. Bold letters with mark “∫” mean no recent national BCG vaccination program for all.

**Table 3 ijerph-17-05589-t003:** Association between mortality as the outcome and morbidity as the exposure, adjusted for number of PCR-tests performed.

Variable	Coef.	Std. Err.	*p*-Value	95% CI	Adjusted R^2^
Morbidity per million population (log10)	1.395	0.083	<0.001	1.232 to 1.558	0.7067
PCR-tests per million population (log10)	−0.241	0.086	0.005	−0.416 to −0.075	

Coef.—Coefficient; Std. Err.—Standard Error; 95% CI—95% confidence interval.

**Table 4 ijerph-17-05589-t004:** Associations between COVID-19 morbidity and mortality per million population and world data for each country.

Variable	*N* * ^3^	Median, IQR	Min (Country) Max (Country)	Morbidity * ^1^ Adjusted R^2^	Mortality * ^2^ Adjusted R^2^
Population, *n*	173	1.02 × 10^7^ 3.5 × 10^6^–3.3 × 10^7^	3.9 × 10^5^ (Monaco) 1.4 × 10^9^ (China)	0.3732	0.7120
Yearly change (%)	173	1.11 0.35–1.96	−1.35 (Lithuania) 3.84 (Niger)	0.3822	0.7172
Net change, *n*	173	89,516 8516–447,563	−383,840 (Japan) 1.40 × 10^7^ (India)	0.3713	0.7073
Population density, *n*/km^2^	173	83 34–167	0 (Greenland) 26,337 (Monaco)	0.3723	0.7048
Land area, km^2^	173	183,630 35,410–581,795	1 (Monaco) 1.6 × 10^7^ (Russia)	0.3726	0.7079
Migrants, *n* (net)	168	−1590 −16,026–14,440.5	−653,249 (Venezuela)) 954,806 (USA)	0.3885	0.7460
Fertility rate, *n*	168	2.27 1.71–3.65	1.11 (Korea) 6.95 (Niger)	0.3784	0.7516
**Urban population (%)**	**168**	**60** **43–78**	**13 (Papua New Guinea)** **98 (Belgium)**	**0.4246 ***	0.7243
World share (%)	173	0.1 0.05–0.4	0 (Saint Lucia, etc.) 18.5 (China)	0.3732	0.7120
Age					
**Median age, years**	**168**	**30** **21.5–39**	**15 (Niger)** **48 (Japan)**	0.3787	**0.7811 ***
0 to 14 years of age (%)	165	26.8 17.6–37.3	12.3 (Niger) 49.8 (Singapore)	0.3789	0.7574
**≥ 60 years of age (%)**	**165**	**9.72** **5.30–20.48**	**2.80 (UAE)** **34.01 (Japan)**	0.3802	**0.7986 ***
**≥ 70 years of age (%)**	**162**	**3.41** **2.03–9.17**	**0.53 (UAE)** **18.49 (Japan)**	0.3818	**0.8027 ***
Economy					
GDP, million US dollars	170	40,729 11,135–223,780	393 (Sao Tome) 1.90 × 10^7^ (USA)	0.3701	0.7095
GDP per capita, US dollars	170	51.4 1534–15,347	104 (Somalia) 165,421 (Monaco)	0.3722	0.7051
Total unemployment rate (%)	164	5.85 3.75–9.40	0.1 (Qatar) 28.2 (Lesotho)	0.3933	0.7436
Male unemployment rate (%)	164	5.5 3–8.5	0 (Qatar) 26 (Eswatini)	0.3905	0.7433
Female unemployment rate (%)	164	7 4–12	0 (Niger) 38 (Syria)	0.4078	0.7435
Total labor force participation rate (%)	164	62 55–68	38 (Yemen) 87 (Qatar)	0.3939	0.7597
Male labor force participation rate (%)	164	73 66.5–79	45 (Moldova) 95 (Qatar)	0.3809	0.7602
Female labor force participation rate (%)	164	53.5 45–60.5	6 (Yemen) 84 (Rwanda)	0.4216	0.7479
National BCG vaccine program					
Annual incidence of tuberculosis, *n*/100,000	164	57 17–176	2 (UAE) 834 (South Africa)	0.3693	0.7053
**Recent BCG vaccine coverage (%)**	**173**	**93** **81–98**	**0 (Italy, etc.)** **99 (Japan, etc.)**	0.3814	**0.7415 ***
Global health observatory					
Health policy					
International Health Regulations score	170	64 48 to 82	17 (Central African Republic) 99 (Canada, Russia)	0.3817	0.7205
**Universal health coverage index**	**166**	**0.41** **0.36 to 0.47**	**0.17 (Saint Lucia)** **0.70 (Peru)**	0.3812	**0.7644 ***
Total expenditure on health as a percentage of gross domestic product	169	6.38 4.76–8.43	1.48 (Timor-Leste) 17.14 (USA)	0.3977	0.7283
Population with household expenditures on health greater than 10% of total household expenditure/income (%)	146	6.56 3.13–12.76	0.20 (Gambia) 54.2 (Sierra Leone)	0.3956	0.7906
Medical personnel					
Hospital beds, *n*/10,000 population	164	21 11–40	1 (Mali) 134 (Japan)	0.3718	0.7409
**Medical doctors, *n*/10,000 population**	**169**	**15.7** **3.3–29.8**	**0.1 (Tanzania)** **84.2 (Cuba)**	0.3800	**0.7351 ***
Nursing and midwifery personnel, *n*/10,000 population	169	26.6 9.4–64.4	0.06 (Cameroon) 201.6 (Monaco)	0.3823	0.7203
Licensed qualified anesthesiologists actively working, *n*	141	250 18–1511	0 (Congo) 194,634 (China)	0.3904	0.6924
Health biomarkers					
High blood pressure (SBP > 140 OR DBP > 90) (crude estimate) (%)	167	23 20–25	13 (Peru) 41 (Croatia)	0.4091	0.7199
Elevated fasting blood glucose (>7.0 mmol/L or on medication) (crude estimate) (%)	167	8.1 5.9–9.7	2.6 (Burundi) 16.6 (Fiji)	0.4069	0.7162
E**levated total cholesterol (≥5.0 mmol/L or 193 mg/dL)** **(crude estimate) (%)**	**167**	**37.6** **25.2–52.5**	**14.8 (Niger)** **69.7 (Denmark)**	0.4070	**0.7401 ***
BMI					
Mean BMI, kg/m^2^	167	26.2 23.4–27.2	20.5 (Ethiopia) 30.0 (Saint Lucia)	0.4233	0.7102
BMI ≥ 30 kg/m^2^ (%)	167	19.9 7.5–25.7	2.1 (Vietnam) 37.3 (USA)	0.4393	0.7135
BMI ≥ 25 kg/m^2^ (%)	167	53.5 27.8–62.2	17.9 (Timor-Leste) 72.1 (Kuwait)	0.4454	0.7204
Alcohol drinking, total consumption per capita among persons aged ≥ 15 years, liters of pure alcohol over a calendar year	168	6.4 2.6–9.8	0 (Bangladesh) 15.2 (Moldova)	0.3834	0.7191
Prevalence of smoking any tobacco product among males aged ≥15 years (%)	119	32 23–44	9 (Ethiopia) 76 (Indonesia)	0.4410	0.8006
Prevalence of smoking any tobacco product among females aged ≥ 15 years (%)	119	8 3–19	0 (Niger) 40 (Serbia)	0.3932	0.8192
**Prevalence of insufficient physical activity among adults aged ≥ 15 years (crude estimate) (%)**	**143**	**28.7** **19.0–36.2**	**5.0 (Uganda)** **65.3 (Kuwait)**	**0.4984 ***	0.7270
Estimated population-based prevalence of depression (%)	166	4.4 4.0–5.0	3.0 (Papua New Guinea) 6.3 (Ukraine)	0.4062	0.7486
Mortality according to age group					
Neonatal mortality rate, *n* per 1000 live births	170	9.6 3.5–21.0	0.9 (Japan) 42.0 (Pakistan)	0.3811	0.7234
Infantile mortality rate, *n* per 1000 live births	170	14.0 5.9–34.9	1.4 (Finland) 84.5 (Central African Republic)	0.3810	0.7507
Under-five mortality rate, probability of dying by age 5/1000 live births	170	14.4 7.0–44.2	1.7 (Finland) 121.5 (Somalia)	0.3813	0.7145
Mortality rate for 5–14-year-olds, *n*/1000 children aged 5–14 years	168	2.8 1.5–8.6	0.4 (Luxembourg) 37.3 (Niger)	0.3781	0.7117
Adult mortality rate, probability of dying between 15 and 60 years of age/1000 population	167	150 96–224	49 (Switzerland) 484 (Lesotho)	0.3812	0.7377
Probability of dying between age 30 and exact age 70 from cardiovascular disease, cancer, diabetes or chronic respiratory disease	166	18.4 14.7–22.6	7.8 (Korea) 30.6 (Yemen)	0.3886	0.7061
Life expectancy at birth, years					
At birth	166	73.4 65.3–81.5	52.9 (Lesotho) 84.2 (Japan)	0.3806	0.7600
At age 60 years	166	19.6 17.2–22.1	13.4 (Sierra Leone) 26.4 (Japan)	0.3838	0.7630
Healthy life expectancy (HALE), years					
At birth	166	65.3 57.5–68.1	44.9 (Central African Republic) 76.2 (Singapore)	0.3795	0.7599
**At age 60 years**	**166**	**14.8** **13.0–17.1**	**10.3 (Sierra Leone)** **21.0 (Singapore)**	0.3825	**0.7652 ***
Disease-specific mortality					
Chronic obstructive pulmonary disease	166	6.13.9–8.3	0.6 (Qatar) 36.8 (India)	0.3975	0.7436
Ischemic heart disease	166	21.5 13.4–35.2	4.9 (Brunei Darussalam) 106.9 (Georgia)	0.3789	0.7431
Lower respiratory infections	166	8.2 3.7–31.6	0.22 (Finland) 130.3 (Chad)	0.3791	0.7455
Stroke	166	10.2 5.6–16.6	1.4 (Qatar) 49.3 (Georgia)	0.3963	0.7726
Tracheal, bronchial, lung cancers	166	1.8 0.8–4.2	0.2 (Niger) 20.5 (China)	0.3971	0.7541
Air pollution					
Ambient and household air pollution attributable death rate /100,000 population	166	65.9 38.5–97.8	8.5 (Georgia) 184 (Brunei Darussalam)	0.3861	0.7427
Concentrations of fine particulate matter (PM_2.5_)	170	21.1 14.6–34.7	5.7 (New Zealand) 94.3 (Nepal)	0.3940	0.7210
Coverage rate with the first dose of a measles-containing-vaccine (MCV1) among one-year-olds (%)	169	93 84–97	30 (Equatorial Guinea) 99 (Mongolia and 23 other countries)	0.3851	0.7095

IQR—interquartile range; GDP—gross domestic product; BMI—body mass index—weight (kg)/height (m)^2^; * ^1^—adjusted for PCR tests per million population; * ^2^—adjusted for PCR tests and COVID-19 morbidity per million population; * ^3^—number of countries the data were able to abstract, *: *p* < 0.001. Bold letters mean statistically significant: *p* < 0.001.

**Table 5 ijerph-17-05589-t005:** Factors associated with morbidity adjusted for PCR-test rates.

Variable	Coef.	Std. Err.	*p*-Value	95% CI	Adjusted R^2^
PCR-tests per million population (log10)	0.574	0.101	<0.001	0.374 to 0.774	0.5037
Urban population (%)	0.764	0.329	0.02	0.113 to 1.416	
Insufficient physical activity (%)	0.015	0.006	0.01	0.004 to 0.026	

Coef.—Coefficient; Std. Err.—Standard Error; 95% CI—95% confidence interval.

**Table 6 ijerph-17-05589-t006:** Factors associated with mortality adjusted for morbidity and PCR-test rates.

Variable	Coef.	Std. Err.	*p*-Value	95% CI	Adjusted R^2^
Morbidity per million population (log10)	1.342	0.067	<0.001	1.210 to 1.475	0.8254
PCR-tests per million population (log10)	−0.485	0.099	<0.001	−0.680 to −0.290	
Population ≥ 60 years of age (%)	0.030	0.010	0.003	0.011 to 0.050	
BCG vaccine coverage	−0.004	0.002	0.006	−0.007 to −0.001	
Universal health coverage	1.473	0.549	0.008	0.387 to 2.560	
Medical doctors/10,000 population	0.003	0.004	0.47	−0.005 to 0.011	
Elevated total cholesterol	0.002	0.006	0.76	−0.010 to 0.014	
HALE at age 60	−0.03	0.029	0.27	−0.090 to 0.025	

HALE—healthy life expectancy, Coef.—Coefficient; Std. Err.—Standard Error; 95% CI—95% confidence interval.

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
