# Peer review of "BCG Vaccination and Mortality of COVID-19 across 173 Countries: An Ecological Study"

_ijerph, 2020, doi:10.3390/ijerph17155589_

Round 1
Reviewer 1 Report
This article described fewer COVID-19 morbidities and mortalities in Bacillus Calmette-Guérin (BCG)-vaccinated countries than BCG-non-vaccinated countries. This paper is very well written.
The reviewer asks one question.
BCG strains include various strains such as early and late strains (e.g. Tokyo 172, Connaught, Tice….). Some papers say that the effect of the vaccine varies depending on the BCG strains. Did this study compare BCG strains? COVID-19 mortalities rates may vary depending on the BCG strains.
Reviewer 2 Report
Very interesting conclusions and suggestion for further research with stronger evidence. Please answer my comments below.
1) Kindly explain why was vaccination uptake assessed as 0% for countries that have stopped their BCG vaccination programs? Did you use any time criterion (e.g. program end date)?
2) "When the number of total deaths was zero, these were changed to 0.01 per million population." - please explain why did you choose value of "0.01"?
Please consider discussing:
1) ... why young age (5-14 years) is an important factor to predict mortality.
2) ... possible causes of differences between morbidity and mortalilty predictors.
Minor:
Line 160 - "0.4 > rho ≥ 0: moderate" - missing ".2"?
Reviewer 3 Report
The authors performed a traditional ecological or correlation study. Unfortunately, the results of these studies generally provide only hypotheses that require further investigation with much more suitable methodologies such as cohort studies or clinical trials.
Therefore, the outcome of this work should be considered high limited with a potential risk of high bias.
Abstract
Line 19: A total of 61 factors were applied.
Line 20: It should be generally reported clinical trials or cohort studies.
Introduction
Line 74: It is not necessary to explain what BMI is.
Material and Methods
2.6. Covariates
In the manuscript, there are evaluated a total of 61 independent variables (factors). Therefore it should be rightly displayed in chapter 2.6. Covariates
2.7.1. Linear regression models
Line 148-149 + Line 158: The authors claimed that the lognormal distribution was achieved. Why the Spearman nonparametric correlation (suitable for non-normal data) was used instead of Pearson parametric correlation, usually used for normally distributed data? It should be explained or correct.
Line 155-157: The authors should use the right terminology. No cut-off point is used but significance level of alfa <0.0001. It should be corrected.
Line 158: The authors should be named R2 as a coefficient of determination.
Line 160: Authors displayed coefficient of correlation as rho. They defined a strength of association. Because coefficient of correlation range between -1 and +1 the evaluation of association is not correct. It should be written an absolute value of rho ≥ 0.5 etc.
Results
Table 1 and 2:
Morbidity and mortality as well as PCR-test should be displayed in person-time, such as person-year or person-month.
Moreover, the rank of the lowest 20 countries is missing in both tables.
3.2. Associations between morbidity, mortality and PCR-tests per million population
Line 215: The authors incorrectly used PCR-testing as a confounder. The ecological studies cannot reveal confounders among factors. So all covariates should be evaluated as potential predictors. Only factors exhibiting collinearities should be considered to be potential confounders.
Line 239: Figure 3 should be complete with legend.
Table 4
It is not necessary to report t-value. Table should be clearer, i.e. p-value could be expressed by GP formatting or APA/NEJM formatting for p-value.
3.4. Screening factors associated with mortality
Mutually adjusted coefficient of correlation resulting from multivariable linear regression exhibited interaction between selected predictors as demonstrated outcomes from table 6 compared to results of table 4. Therefore, multivariable regression with more than 2 or 3 predictors contributes to more valuable results. It is clear that extremely large of covariates were selected for such a limited set of 172 countries. This sample size could allow to use maximum of 10 covariates. This point of view should be discussed.
3.5. Gradient boosting
Line 315 + 320: The mean of coefficient of determination should by specified, i.e. for morbidity, or mortality.
Discussion
Line 359: The ecological study is also observational one. Therefore, it should be better specified, such as cohort study, etc.
The main limitation of this study was absence of anti-epidemic measures. They generally play the significant role in epidemic, i.e. range of measures, their timely implementation etc. Therefore, this present study results are burdened with an extreme error.
Reviewer 4 Report
This is an interesting study on a timely topic, in which the authors investigate whether national BCG vaccination programs can lower SARS-CoV-2 mortality and/or mortality rates. In addition the authors test out a novel machine learning approach to their data analysis.
A strength of this paper is the large number of covariates and datasets that were included in the study, and the unbiased machine learning approach.
However, there are a few important questions that should still be addressed:
- are there countries that have stopped a national BCG program in the last 50 years? if so, there may still be a protective effect in the at-risk >60-years-of-age population. perhaps this could be looked at as well as a separate piece of information.
- are there other national vaccine programs that the BCG data could be compared to? this is important to show the effect is specific to the BCG and not due to a superior vaccination system overall, or other unknown factors that are somehow linked to BCG.
- could you add a column or a symbol to Tables 2 and 3 indicate which of these countries have a BCG vaccination program?
Reviewer 5 Report
The present study by Urashima et al. reported a hot issue on COVID-19 that we all experience day to day life from the beginning of the pandemic and still, the infection is in the full flag and several papers and comments and facts are discussed. The MS is interesting and putting evidence on BCG vaccination and mortality of COVID-19 cases across 173 counties. I agree with the statement of authors that there should be needed for adjusting potential confounding factors to check any relationship between mortality and BCG vaccination.
The authors conducted an ecological study using data from 173 countries, from open sources the study choose 58 factors as well as BCG vaccine coverage (%) that significantly associated with COVID-19 morbidities and mortalities by using a regression model. The study found that ‘Urban population (%)’ and ‘insufficient physical activity (%)’ in each country was positively related to morbidity, but not mortality, after adjustment for PCR-tests. However, BCG vaccine coverage (%) was negatively associated with mortality. Collectively the study support that a national BCG vaccination program seems to reduce mortality of COVID-19, however, the study needed to be proved by randomized clinical trials.
Currently, it is difficult to say that greater BCG vaccine coverage reduces the risk of deaths due to COVID-19. The current situation is alarming. We need more time to see the situation and put our efforts into the matter. After going through your paper, I have mention comments that need to be responded.
- Line 25-27, The line needs to be rewritten as the proper meaning was not clear
- Line 38-40, better to add more recent papers and information related to your study
- Line 49-54, Big paragraph, need to divide it two sentences
- Line 57-58, now you can update more information
- Line 61, several scientists proposed a hypothesis [7] the author had mentioned only one reference
- Line 73, It is better to explain here for using a linear regression model just for the clarity.
- the author needs to explain the utility of gradient boosting in his study
- Line 134-136, I saw some parameters that are closely related to each other, how you categorize those patients who already come in one group and the others.. Such as you see in covariate 46 and 47, the same patients can come in both.. because distinguish is very important.
- Line 150-151, The proper meaning of the sentence was not clear. please rewrite it.
- Line 157, better not to use the cut-off point for the significance level
- Line 172, it is important to discuss this method in a simpler way so that the importance of the method can be highlighted. There are other tools of machine learning why the author chooses this one.
- In the method section Line 173, I think it is better to explain in a good representation such as by sketch or ray diagram.
- For table 2, I could not figure it out as the information needs to be updated in a timely manner.
- Line 210-211, please elaborate on what does it indicate.
- Line 229, Higher morbidity but fewer PCR-tests were associated with higher mortality please elaborate this line for the better understanding
- Line 249-250, here only the author should write about the information related to mortality.
- Table 4, the number of countries show 172
- Figure 5, why only 143 countries and it seems that it was stronger than the above
- Table 6, why there has been so much change in the t value
- Line 307, need to be check once again if possible (with the current findings)
- Line 315-318, What does it indicate
- Related to the discussion of this paper: need to be rewritten, as it required changes and the author should include a paragraph on countries that deal with a good policymaking in reducing the morbidity and mortality of COVID
- Line 332, possible there was some missing information here
- Line 336, need a reference, I think now the author can find some published paper
- The paper has insufficient references based on the cited in the paper.
Round 2
Reviewer 4 Report
Thank you for your responses and your hard work editing the manuscript.
Reviewer 5 Report
The author addressed all concerns.
The paper is good enough to publish in the IJERPH.